# Probing hydrogen bond strength in liquid water by resonant inelastic X-ray scattering

Vinícius Vaz da Cruz [1], Faris Gel'mukhanov[1,2], Sebastian Eckert [3], Marcella Iannuzzi [4], Emelie Ertan[5], Annette Pietzsch [6], Rafael C. Couto[1], Johannes Niskanen[6,7], Mattis Fondell [6], Marcus Dantz[8], Thorsten Schmitt[8], Xingye Lu[8], Daniel McNally[8], Raphael M. Jay [3], Victor Kimberg [1,2], Alexander Föhlisch [3,6] & Michael Odelius [5]

Local probes of the electronic ground state are essential for understanding hydrogen bonding in aqueous environments. When tuned to the dissociative core-excited state at the O1s pre-edge of water, resonant inelastic X-ray scattering back to the electronic ground state exhibits a long vibrational progression due to ultrafast nuclear dynamics. We show how the coherent evolution of the OH bonds around the core-excited oxygen provides access to high vibrational levels in liquid water. The OH bonds stretch into the long-range part of the potential energy curve, which makes the X-ray probe more sensitive than infra-red spectroscopy to the local environment. We exploit this property to effectively probe hydrogen bond strength via the distribution of intramolecular OH potentials derived from measurements. In contrast, the dynamical splitting in the spectral feature of the lowest valence-excited state arises from the short-range part of the OH potential curve and is rather insensitive to hydrogen bonding.

[1] Theoretical Chemistry and Biology, Royal Institute of Technology, 10691 Stockholm, Sweden. [2] Laboratory for Nonlinear Optics and Spectroscopy, Siberian Federal University, 660041 Krasnoyarsk, Russia. [3] Institut für Physik und Astronomie, Universität Potsdam, Karl-Liebknecht-Strasse 24-25, 14476 Potsdam, Germany. [4] Physical Chemistry Institute, University of Zürich, 8057 Zürich, Switzerland. [5] Department of Physics, Stockholm University, AlbaNova University Center, 10691 Stockholm, Sweden. [6] Institute for Methods and Instrumentation in Synchrotron Radiation Research FG-ISRR, Helmholtz-Zentrum Berlin für Materialien und Energie, Albert-Einstein-Strasse 15, 12489 Berlin, Germany. [7] Department of Physics and Astronomy, University of Turku, FI-20014 Turun yliopisto, Finland. [8] Photon Science Division, Paul Scherrer Institut, CH-5232 Villigen PSI, Switzerland. Correspondence and requests for materials should be addressed to V.V.d.C. (email: vvdc@kth.se) or to M.O. (email: odelius@fysik.su.se)

Hydrogen bonding in aqueous solutions influences a vast array of processes in chemistry, biology and atmospheric science. Vibrational infra-red (IR) spectroscopy is an established technique for investigations of hydrogen bonding, which induces a systematic red-shift of the OH vibrational frequency. Complemented by theoretical simulations, it has been used to study variations in the local hydrogen bond (HB) environment in liquid water[1,2]. The spectral shape and inhomogeneous broadening in the IR spectrum originate not only from inhomogeneities in the HB configurations, but also from intra- and inter-molecular couplings. Therefore, diluted isotope substitution is regularly employed to enhance the sensitivity to HB environment by localising the OH and OD chromophores[1]. In addition, time-resolved pump-probe and multi-dimensional correlation spectroscopy using short IR pulses[1–3] provide insights into the structural dynamics of the HB network and the dynamics of vibrational energy redistribution. These methods have been used to derive information of correlation or dephasing time and life-times, which have clarified the influence of hydrogen bonding on the IR spectrum. The IR intensity enhancement for hydrogen-bonded configurations makes IR absorption and Raman spectroscopies very useful probes of the HB network[2,4,5].

High-resolution resonant inelastic X-ray scattering (RIXS) offers a complement to IR vibrational spectroscopy. However, how sensitive RIXS and X-ray fluorescence are to HB rearrangements, and the underlying structural implications, is a hotly debated[6–15] topic. Earlier, the long vibrational progression in quasi-elastic RIXS has been empirically analysed; Either attributed to highly weakened/broken donating HBs selected by the pre-edge core-excitation in the framework of a single bond approximation[16]. Or assigned to symmetric and anti-symmetric normal modes and to OH vibrations in a broken-bond molecule[11]. However, a water molecule has in general two non-equivalent OH bonds, due to the asymmetric surroundings. This necessitates a strict coherent treatment of both oscillators as carried out in our simulations presented below. Furthermore, the need for analysing RIXS with a quantitative theoretical framework has been demonstrated by the recent meticulous investigations of gas-phase water[17–19]. Based on quantum dynamical simulations of the nuclear wave packet, details in the RIXS spectrum of the water molecule have been explained in terms of the shape of the potential energy surfaces[17–19]. In particular, the ground state potential energy curves (PECs) in gas-phase water were recently extracted in different directions by varying the excitation energy to scatter against different core-excited states[20]. In contrast, the concept of a unique local potential energy surface is not applicable to liquid-phase because of the fluctuating HB network.

There are competing conceptions of the local structure of liquid water; Either as a continuum of different HB configurations[21–23], or as a mixture of two structural motifs[6,12]. Additionally, the partial contributions of different structures to the X-ray absorption (XAS) and IR spectra as well as the average number of hydrogen bonds per molecule in liquid-phase are subjects of discussion[1,15,23–26]. There is an intrinsic problem with such analyses because the relative weights of different structures are strongly dependent on the associated transition dipole moments[26] and transition energies, both of which are derived from theory, typically with limited accuracy[27] due to the large system size. Admittedly, the problem is more serious in X-ray than in IR spectroscopy because of the lower accuracy of calculation of highly excited states. Furthermore, evaluation of conceptual models of broken and intact HBs is complicated, since each experimental probe is sensitive to certain aspects of the HB environment.

RIXS channels decaying into valence-excited states are often broadened by variations in valence-excitation energy for different environments[28], but may still exhibit sharp features of core-excited state dynamics just as in gas-phase[7,8,10,19]. Hence, there have also been attempts[6,12,13] to investigate the local structure in liquid water using X-ray fluorescence decay channels, specifically the ones involving a transition between the non-bonding $1b_1$ lone-pair and the $1s_O$ core-hole, for which a splitting of the spectral feature is observed. Although a molecular mechanism for the splitting was not explicitly given, the role of nuclear dynamics has been established by experimental observations[7] of an isotope effect of this transition (see also refs. [8,9,11]).

In this study, we concentrate on the sensitivity of RIXS to the local structure from the perspective of the local potential energy surface. We do so by exploring the local variation of the potential energy landscape in the ground state of liquid water directly from RIXS measurements by using an approach based on quantum-classical simulations of RIXS. The theoretical framework is composed of classical ab initio molecular dynamics (MD) simulations, calculation of local potential energy surfaces from the sampled configurations, and quantum wave packet modelling of the nuclear motion in relevant degrees of freedom (see Methods). Thereby, we reach insights into the variations in the local HB environment, which strongly affects the long-range part of the OH PEC. For enhanced insight, we derive the distribution of PECs of OH bonds with intact and broken HBs as reconstructed from experimental RIXS data. The method of reconstruction is inspired by the observed breakdown of the one-to-one correspondence between RIXS peak positions and vibrational quantum numbers. In contrast by analysis of the dynamic mechanisms, we show that the splitting, emerging for pre-edge core-excitation, has a purely dynamical origin and is primarily sensitive to the short-range part of the PEC since the splitting is formed at short time-scales before fragmentation. Altogether, we established how different RIXS channels deliver separate information; about the local structure via long-range dynamics in quasi-elastic RIXS and about short-range dynamics, which is much less sensitive to the structure, in the electronically inelastic $1b_1$ channel.

## Results

**Gas vs. liquid-phase**. Vibrationally resolved RIXS measurements of gas-phase and liquid water, presented here (Fig. 1a, b), were performed at the Swiss Light Source[29] (see Methods). The photon frequency $\omega$ was tuned near resonance with the lowest core-excited state and the decay back to the ground electronic state was studied as a function of energy loss $\omega - \omega'$, where $\omega'$ is the frequency of the emitted photon. In gas-phase, core-excitation to the $\left|1s_O^{-1}4a_1^1\right\rangle$ state leads to ultra-fast dissociation along the OH bonds[17–19] (Fig. 2a). The propagation of the nuclear wave packet[17–19] results in the long vibrational progression seen in both theory and experiment. In the liquid, however, we observe a strong shortening of the vibrational progression in comparison to the gas-phase (Fig. 1a, b). Our simulations show that this shortening arises from variations in the OH PECs, reflecting the different local HB environments (Fig. 1c) in liquid water. These variations affect mainly the long-range part of the OH PEC and result in a variation of the high vibrational levels, seen in the partial density of vibrational states of the $n$-th group $\rho_n(\epsilon)$ (see Fig. 1d and Methods). Each group is characterised by the group number $n = n_1 + n_2$[17], where $n_1$ and $n_2$ are the vibrational quantum numbers for the stretching modes along the OH bonds (see Methods). In Fig. 1d, we notice for $n \geq 2$ a strong overlap of the partial density of states $\rho_n$ belonging to different groups.

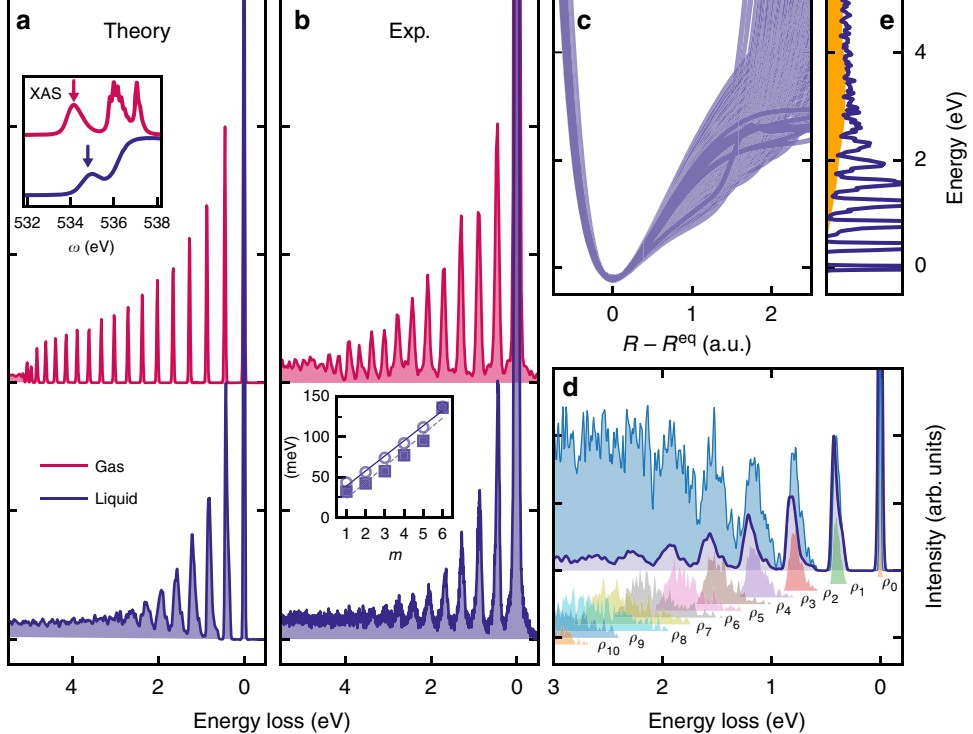

**Fig. 1** Resonant inelastic X-ray scattering (RIXS) spectra of water in gas/liquid-phase under $4a_1$/pre-edge core-excitation. **a** Theoretical RIXS spectra of gas-phase and liquid water vs. the energy loss ($\omega - \omega'$). The inset shows energy of resonant excitation in the experimental XAS of free water molecules and liquid water (from ref. [15]). **b** Experimental RIXS spectra of gas-phase and liquid water. The inset compares experimental (circles) and theoretical (squares) peak widths in meV as a function of the peak number $m$. **c** Ab initio potential energy curves along the OH bonds for each of the 64 sampled water molecules in liquid water. **d** Partial densities of the vibrational states $\rho_n$ (see Eq. (7)) and the total density of states $\rho = \sum_n \rho_n$ together with the RIXS profile $\sigma$. **e** Overlap of the partial RIXS cross-sections results in the formation of a background shown in yellow (compare with the overlap of the partial densities of vibrational states **d**)

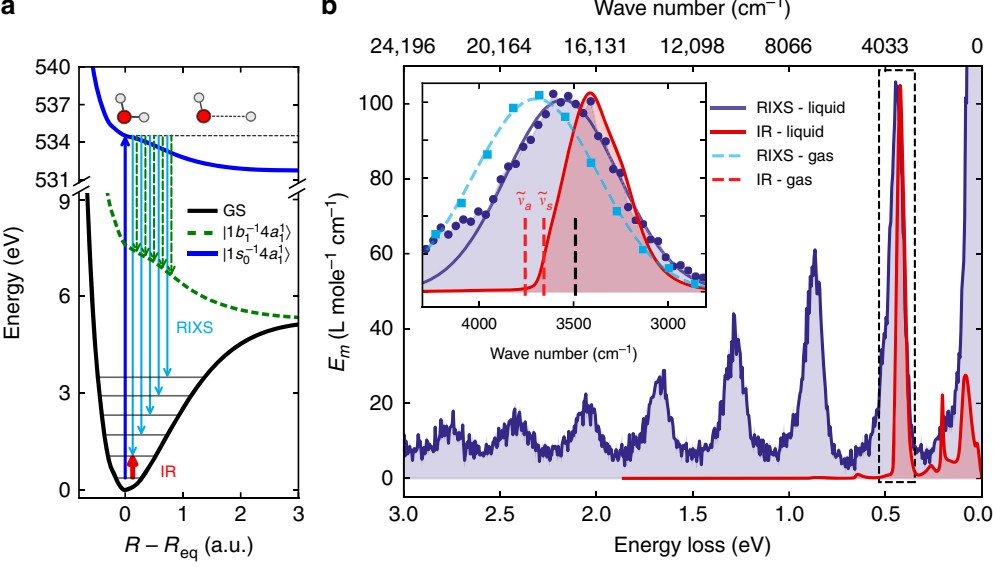

**Fig. 2** Comparison of resonant inelastic X-ray scattering (RIXS) under pre-edge core-excitations and infrared (IR) spectra of water in gas and liquid phases. **a** Schematic representation of the RIXS transitions in terms of the gas-phase PECs, for better visualisation the vibrational levels are only qualitatively depicted. **b** Direct comparison of RIXS and IR[30] data for liquid water. The inset shows the first vibrationally excited peak in RIXS in both gas- and liquid-phase. The lines depict a gaussian fit of the measured data points (circles and squares). The dashed red bars show the experimental positions of the gas-phase symmetric ($\nu_s$) and anti-symmetric ($\nu_a$) stretching frequencies, while the black dashed bar shows the maximum position of the theoretical density of states computed by Auer and Skinner[4]

Hence, the adjacent density of states $\rho_{m-1}(\epsilon)$ and $\rho_{m+1}(\epsilon)$ can contribute to the $m$-th peak. The high-energy part of the spectrum ($\gtrsim 3$ eV) is smeared (Fig. 1a, b) by variations in local environment into a smooth background (Fig. 1e). Thus, the overlap of the partial density of states $\rho_n$, and the related increase in peak width (see inset in Fig. 1b), qualitatively explains the shortening of the spectrum.

As mentioned above, RIXS provides a vibrational probe complementary to IR spectroscopy. Therefore, it is pertinent to compare the RIXS and IR spectra of water (Fig. 2). Contrary to RIXS, the main contribution in IR absorption originates from the $0 \rightarrow 1$ dipole allowed OH transition (higher lying dipole forbidden IR transitions are more than two orders of magnitude smaller[30]). Hence, the majority of IR studies of liquid water[1,2] has focused only on this transition, which probes the bottom of the OH potential well. One can see that the liquid–gas shift in IR absorption ($\approx 280$ cm$^{-1}$) is significantly larger than in RIXS ($\approx 140$ cm$^{-1}$) and the RIXS and IR peaks are shifted in opposite directions with respect to the theoretically derived maximum of the OH vibrational density of states[4] at $\approx 3490$ cm$^{-1}$ (Fig. 2b).

The sensitivity of IR spectroscopy to a large extent stems from the strong dependence of the IR intensity of the OH stretch on hydrogen bond environment[2,4,5]. The IR absorption transition dipoles of the OH stretching modes with a broken HB (located in the high-frequency region) are significantly smaller than those of hydrogen-bonded OH modes (located in the low-frequency region)[5]. The situation is reversed in RIXS, where the molecules with a weak/broken hydrogen bond are excited preferentially. This explains the opposite shifts of the RIXS and IR absorption resonances with respect to the maximum of the density of vibrational states (see inset in Fig. 2b). Thus, IR and RIXS spectroscopy complement each other and deliver structural information already at the lowest $0 \rightarrow 1$ transition: this OH transition in IR absorption evidences the existence of structures with strong HB contrary to RIXS where the peak position of the $0 \rightarrow 1$ transition is associated with both broken and strong HBs structures (see next Section and Fig. 3c). One should mention that IR Raman spectra[2,4,5] show a similar trend as the IR absorption spectra.

There is another effect intrinsic to liquid-phase, namely motional narrowing caused by the fluctuations in the environment[1,31], which can affect the spectral shape of individual vibrational peaks of quasi-elastic RIXS similarly to IR absorption, since both RIXS and IR end up in the same final state. This dynamical effect is neglected here because we use a static environment (see Methods). To justify the static approximation, it is worth noting that the broadening of a vibrational resonance has two representative limits defined by the dimensionless parameter $\Delta\omega\tau_c$, where $\Delta\omega$ is the variance of frequency fluctuation in the liquid, while $\tau_c$ is the decay time of the frequency fluctuation correlation function[32]. In the regime of slow modulation (static regime) $\Delta\omega\tau_c \gg 1$ the line width is large and is given by inhomogeneous broadening. The regime of motional narrowing occurs in the opposite case of fast modulation $\Delta\omega\tau_c \ll 1$ in which the line width is defined by the homogeneous broadening. According to two-dimensional (2D) IR spectroscopy $\tau_c \approx 176$ fs for water[33]. MD simulations[31] have shown that bath fluctuations reduce the spectral width of the main OH IR peak ($n = 1$) by 30% for a value of $\Delta\omega\tau_c \sim 1$. Both experiment and simulations (Fig. 1b) show that $\Delta\omega$ grows rapidly on the way to higher vibrational resonances (main focus of our study) where $\Delta\omega\tau_c > 1$ and the regime of motional narrowing is switched to the static regime.

**Role of asymmetric bonds**. It has been argued that oxygen K-edge X-ray spectra are sensitive to the local structure of the liquid

water[6,12,21,22,24]. Here, we investigate the sensitivity of RIXS to hydrogen bonding based on the classification of "double-donor" (D2) and "single-donor" (D1) structures (see Supplementary Fig. 1), in which either both OH groups, or just one OH group, in the water molecule donate a hydrogen bond (see Methods). The pre-edge region in the XAS of liquid water has been ascribed to excitation of molecules in asymmetric HB environments (D1), where the assumed selectivity depends on the XAS simulation method[24,34–36]. The employed excited core-hole (XCH) approximation[35,37] yields enhanced transition dipole moments for D1 structures. Hence, the partial RIXS cross-sections $\sigma_{D1}$ and $\sigma_{D2}$ in Fig. 3b, c contribute almost equally to the RIXS profile, even though the D1 structures are in minority (only 20%) in our MD simulation (see Methods). The progression in $\sigma_{D2}$ is red-shifted with respect to $\sigma_{D1}$, since the D2 structures on average experience shallower potentials than the D1 structures. This red-shift together with their intrinsic spread allow us to explain the formation of a background in the total profile $\sigma = \sigma_{D1} + \sigma_{D2}$ (see Figs. 1e and 3d).

In liquid water, vibrational modes are usually localised on the OH bonds due to the asymmetry of the local environment. The first impression is that these localised modes can be treated independently within single-bond approximation. However, because of the shared core-excitation, both OH bonds are coherently excited with the same transition dipole moment in the course of RIXS as illustrated by the nuclear wave packet in Fig. 3a. Even a molecule with one broken HB has a strong hydrogen-bonded OH stretch associated with it. As a consequence, the first RIXS peak of the D1 structures in Fig. 1c bears the signatures of both intact and broken HBs and is broader than for the D2 structures. Thus, our ab initio RIXS analysis taking into account both OH stretches does not support either a single-bond model or the related earlier empirical interpretations based only on the OH stretch with a broken HB[16] (see Supplementary Fig. 2 and dashed profile in Fig. 3c).

The simulated RIXS spectrum of an individual D1 configuration (Fig. 3d) differs very much from the one in gas-phase (Fig. 1). The asymmetric environment leads to a complicated-spectrum formed by single-bond excitations ($(n_1, 0)$ and $(0, n_2)$) and mixed bond excitations ($n_1, n_2$). We see from Fig. 3e that the eigenvalues of the steep potential (weak HB) approximately match the peak positions of the total sampled RIXS spectrum in contrast to the shallow potential (strong HB), which shows considerable deviations as even two eigenvalues may belong to the same peak (Fig. 3d). The neglect of this effect results in an artificially narrow distribution of OH potentials (see below and Fig. 4a) not reproducing the broad distribution of the OH PECs of liquid water seen in Fig. 3e. Our solution of the inverse problem of reconstruction of fluctuating local PECs captures the observed breakdown of the one-to-one correspondence (assumed in refs. [11,16]) between RIXS peak positions and vibrational quantum numbers.

**Confidence intervals for OH potentials**. A key insight from the present simulations is that to extract potential information, we need to design a method that allows for more than one vibrational eigenvalue to be located within the energy range around a given $m$-th peak in the observed RIXS progression. In a conservative attempt to estimate the confidence intervals (see Supplementary Fig. 3), we define this energy range $\Delta\varepsilon_m$ as the spacing between two adjacent minima of this peak and introduce the distribution of the number of vibrational eigenvalues per peak $k_m$ ordered according to the peak's number ($k_1 k_2 k_3 k_4 k_5 k_6$). For

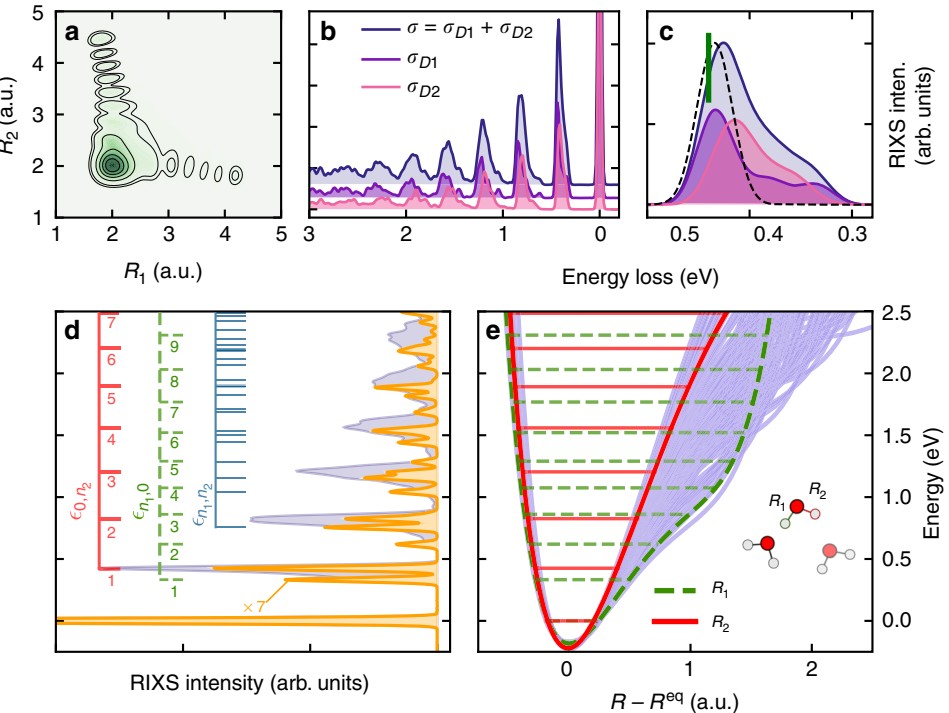

**Fig. 3** Role of coherent excitation of both OH bonds. **a** The nuclear core-excited wave packet squared shows coherent excitation of both OH bonds in a D2 configuration. **b** Partial resonant inelastic X-ray scattering (RIXS) cross-sections $\sigma_{D1}$ and $\sigma_{D2}$. **c** Partial contributions to the RIXS peak $m = 1$. The dashed curve shows the result from the single bond approximation $\sigma_{D1}^{sb}$ simulated including only the single OH stretches with a broken HB. The vertical green bar shows the theoretical peak position of the RIXS in gas-phase water. Current DFT (BLYP) description gives a slight red-shift of the peaks relative to experiment and to high-level calculations[17–19]. **d** Total RIXS profile (blue) and assignment of RIXS spectrum (yellow) of a single asymmetric D1 structure. **e** All OH potentials (blue) of the sampled configurations. Solid red and dashed green curves show the steep (no HB) and shallow (HB) OH potentials of the D1 structure in **d**. Subsets of eigenvalues $\epsilon_{n_1,0}$ (solid red) and $\epsilon_{0,n_2}$ (dashed green) in **d**, **e** assign the single bond excitations associated with potentials of corresponding colour while the subset $\epsilon_{n_1,n_2}$ (blue) displays the mixed excitation overtones

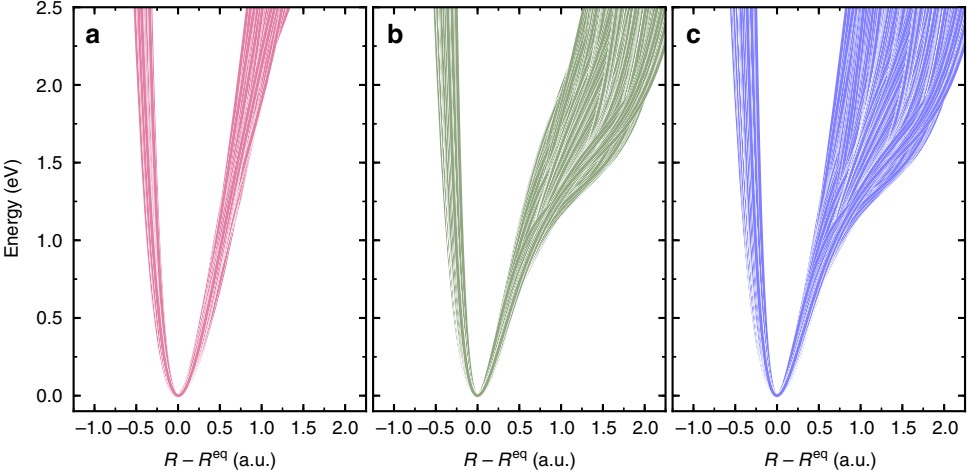

**Fig. 4** Confidence intervals for OH potentials extracted from the experimental resonant inelastic X-ray scattering (RIXS) spectrum. **a** Confidence interval for the PECs in the case of weak HB, corresponding to the constraint in Eq. (1); **b** Confidence interval for PECs of OH bonds in the case of stronger HB (Eq. (2)). **c** The whole set of the PECs obtained by combining **a**, **b**

example

$$\text{weak or no HB} \quad (111111), \quad (111112), \qquad (1)$$

$$\text{strong HB} \quad (112212), \quad (112222), \cdots \qquad (2)$$

Our simulations show that no more than two vibrational

eigenstates can belong to a given peak in the spectrum. The variation of the local environment leads to a variation in the eigenvalue distribution, which depends on the shape of the OH PEC: There is only one vibrational level within each peak for the steep potentials (weak or no-HB), the only exception being the peaks with $m \geq 6$ (see Eq. (1) and Fig. 3d). In contrast, two vibrational levels of a shallow potential (HB) can lie within the

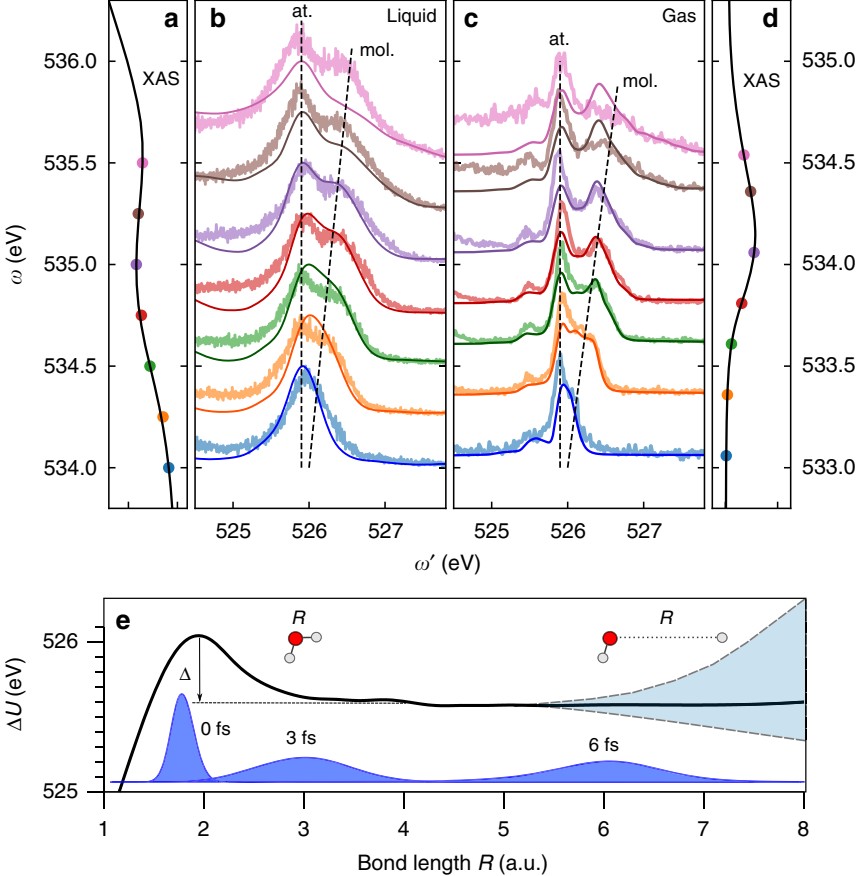

**Fig. 5** Dynamical origin of the splitting of the $1b_1$ peak. Comparison of the dispersion of the split components in experimental RIXS spectra of liquid (**b**) and gas-phase water (**c**) at the pre-edge region. The panels **a**, **d** indicate the corresponding excitation energies in the XAS pre-edge region for liquid and gas-phase water. **c** includes theoretical gas-phase RIXS spectra (solid lines). Solid lines in the **b** show liquid spectra $\sigma_{liquid}(\omega', \omega)$ calculated by convolution of the experimental gas-phase spectra $\sigma_{gas}(\omega', \omega)$ with the structure function $\rho(\omega'_1 - \omega')$ (see Eq. (8)) with $\gamma$(FWHM) = 0.35 eV. Both liquid (**b**) and gas-phase (**c**) spectra display a nondispersive component (pseudo-atomic peak) and a molecular band following the Raman dispersion law. **e** illustrates schematically how the pseudo-atomic peak is formed near equilibrium ($R = 3$ a.u.) as the PECs of the core-excited $U_c(R)$ and final $U_f(R)$ states become almost parallel: $\Delta U = U_c(R) - U_f(R) \approx$ const. The parameter $\Delta \approx 0.45$ eV is the splitting between molecular and pseudo-atomic peaks on top of the XAS resonance

$m$-th peak, except for $m = 1$ (see Eq. (2) and Fig. 3d). Since the eigenvalues belonging to the $m$-th peak are only confined to a $\Delta\varepsilon_m$ interval, the constraints (Eqs. (1) and (2)) generate a distribution of PECs, which defines the confidence interval of the OH potentials. To extract the confidence interval we designed a procedure (see Methods) using a genetic algorithm[38] (see Supplementary Fig. 4), which was validated (See Supplementary Fig. 5) for the theoretical RIXS spectrum of liquid water (Fig. 1a) and then applied to the experimental RIXS spectrum (Fig. 1b) to extract a distribution of PECs in liquid water with a minimal model dependence. The constraint given by Eq. (1) alone results in a narrow confidence interval (Fig. 4a), which is associated solely with the OH potentials weakly affected by HB (steep potentials). In contrast, the constraint defined in Eq. (2) results in a much wider distribution (Fig. 4b) related to various HB configurations. The total reconstructed confidence interval (Fig. 4c) comprises both steep and shallow PECs. Thus, in spite of the inherent chaotic nature of liquid water, RIXS data provides the possibility to, within confidence intervals, separately determine the behaviour of PECs for OH groups involved in weak and strong HBs.

**Dynamical origin of the $1b_1$ splitting**. The local HB environment has also been probed in electronically inelastic processes[6-12];

Decay channels in which the $1s_O$ core-hole is filled by a transition from the occupied lone-pair orbital $1b_1$. This transition forms a split peak (Fig. 5), which has been attributed either to two distinct ground state structural motifs[6,12] or to nuclear motion after core-excitation[7-11]. In spite of experimental evidence of the dynamical nature of discussed splitting, there has been no microscopic explanation of the mechanism of this phenomenon. This motivates us to perform the analysis of the problem in terms of OH PECs and quantum wave packet propagation. Here, we investigate the emergence of the splitting at the pre-edge resonance by looking at the evolution of the RIXS spectrum as a function of photon energy detuning $\Omega$ from the top of the pre-edge peak. The extent of nuclear dynamics can be controlled via the effective scattering duration time[39] $\tau = 1/\sqrt{\Omega^2 + \Gamma^2}$, which reaches a maximum $\Gamma^{-1} = 8$ fs at the resonance, $\Omega = 0$.

The experimental RIXS spectra shown in Fig. 5 display a striking quantitative coincidence of the $\omega$-dependence of the $1b_1$ splitting in liquid and gas phases. This is a clear indication that the splitting is of the same dynamical origin as recently established for the gas-phase[19] associated with the different dispersion laws of the pseudo-atomic and molecular peaks. The pseudo-atomic peak is formed due to the similar non-bonding characters of $1s_O$ and lone-pair $1b_1$ orbitals, which make the potential surfaces of the core-excited and final states almost

parallel already at moderate distortions (see Fig. 5e). Figure 5c shows some disagreement between theory and experiment for the gas-phase RIXS spectra. This is due to the limited accuracy of the calculated OH potentials for the core-excited state and the final states. Environment-dependent fluctuations in emission energy near equilibrium can be characterised by a distribution function $\rho(\omega_1' - \omega')$. Thus, we can approximately reconstruct the liquid spectrum $\sigma_{\text{liquid}}(\omega', \omega)$ by convolution of the experimental gas-phase spectrum $\sigma_{\text{gas}}(\omega', \omega)$ with $\rho(\omega_1' - \omega')$ (see Methods). In Fig. 5, the reconstructed spectrum $\sigma_{\text{liquid}}(\omega', \omega)$ is shown to be in good agreement with the experimental spectrum of liquid water. The employed convolution ignores slight variation of the short-range part of OH potentials for different structures. This is the main reason for the remaining disagreement between theory and experiment seen in Fig. 5b.

Despite that the detailed mechanism of the discussed splitting in the main- and post-edge regions (and for non-resonant core-excitations) is beyond the scope of our article, it deserves a special comment. According to DFT-based MD simulations, the doublet in the region of the $1b_1$ peak for non-resonant excitations is related to the $3a_1$ and $1b_1$ levels, which approach each other in the course of the OH bond elongation in the core-ionised state, which as confirmed by the simulations is dissociative in the local hydrogen bond environment in liquid water[8,9]. It is important to notice that this splitting for higher excitation energies is absent in the gas-phase[10] where (contrary to liquid-phase) the OH potential is bound[9].

To conclude, since both molecular and pseudo-atomic $1b_1$ peaks arise from decay near the equilibrium, the splitting under $4a_1$ core-excitation is not caused by different HB environments (see also ref. [8]) and can not be used as fingerprint of specific local structure.

## Discussion

In conclusion, we present a comprehensive ab initio analysis of the vibrational RIXS spectrum of water and show that the observed progression arises from coherent excitation of both OH bonds of a water molecule embedded in different local environments during the scattering process. Our results indicate that a broad distribution of different configurations and hence potentials contribute to quasi-elastic RIXS at the O1s X-ray absorption pre-edge instead of a very narrow one as previously suggested[16]. Fluctuation of the OH potentials with strong HBs results in large variations in the energy of highly excited vibrational levels, hence leading to broadening of the peaks and ultimately to the shortening of the vibrational progression of RIXS in liquid water with respect to the gas-phase. The distribution of PECs for OH bonds with weak and strong HB have been derived from experimental RIXS to characterise HB strength. Weak HBs lead to a narrower distribution of OH potentials, while strong HBs lead to a much broader distribution. We describe the molecular mechanism, which evidences that the lone-pair ($1b_1$) peak splitting is of dynamical origin, corroborating previous experimental observations of the large isotope effect on the $1b_1$ peak[7,10] and contradicting a structural interpretation[6,12] of the splitting.

## Methods

**Experiment**. The experimental RIXS spectra presented here were measured with the SAXES spectrometer[29] at the RIXS end station of the ADRESS beam line[40] at the Swiss Light Source at the Paul Scherrer Institut. We utilised a flow-cell separating the sample from the vacuum by a $Si_3N_4$ window of 150 nm thickness with a ~10 nm Au coating. The energy calibration was based on the $O_2$ RIXS spectrum[41]. Due to breakdown of the windows in irradiation, the cell was moved every 10 min. To avoid errors from this procedure, the spectra of these individual scans were shifted to same energy scale by using a fit to the elastic line before joining them to a single one for further data processing. The experimental quasi-elastic RIXS spectrum of liquid water[15] for excitation energy tuned on the pre-edge

peak ($\omega \approx 535$ eV) is compared with the gas-phase spectrum excited on the $4a_1$ resonance ($\omega \approx 534.1$ eV[19]) in Figs. 1 and 2. The resonantly scattered photons were detected at 90° angle from the incoming photons with a combined experimental resolution of 40 meV for liquid water and 75 meV for gas-phase water, respectively. The experimental data shown in Fig. 5 for liquid-phase are new except two spectra for $\omega = 534$ and 535 eV taken from ref. [15]. These data are compared with the gas-phase spectra[19].

**Hybrid quantum-classical theory of RIXS**. The simulations of the RIXS spectrum of liquid water were carried out employing a two level classical-quantum approach. First, the liquid-phase was simulated using ab initio MD of a periodic 64-molecule system (as described below). On the second step, cuts through the ground and core-excited potential energy surfaces along both OH bonds were sampled over all 64 water molecules in a snapshot from the MD simulation. These potentials energy curves were used in quantum simulations of the partial RIXS cross-sections $\sigma_k(\omega, \omega')$ for each $k$-th molecule in the configuration. The total RIXS cross-section of the scattering from the ground state (0) via the core-excited state (c) to the final electronic state (f) was calculated as the sum over these partial contributions

$$\sigma(\omega', \omega) = \sum_{k=1}^{64} \sigma_k(\omega, \omega'). \tag{3}$$

In order to compute the vibrationally resolved RIXS, we use a quantum description of the OH vibrations in liquid water. The Hamiltonian, in valence coordinates, on the electronic state $i = 0, c, f$ for each molecule

$$h_k^i = -\frac{1}{2\mu}\left(\partial_{R_1}^2 + \partial_{R_2}^2\right) - \frac{\cos\theta_0}{m_O}\partial_{R_1 R_2}^2 + V_k^i(R_1, R_2), \tag{4}$$

is approximated by assuming an independent bond approximation, $V_k^i(R_1, R_2) \approx V_k^i(R_1, R_2^{\text{eq}}) + V_k^i(R_1^{\text{eq}}, R_2) - V_k^i(R_1^{\text{eq}}, R_2^{\text{eq}})$, with a frozen local environment, where $R_1$ and $R_2$ are the OH bond lengths of the $k$-th molecule; the label (eq) marks the equilibrium position; $\mu = m_H m_O/(m_H + m_O)$ where $m_H$ and $m_O$ are the masses of the hydrogen and oxygen atoms; $\theta_0$ is the equilibrium ∠HOH angle; and $V_k^i$ is the molecular potential along a OH bond of the $k$-th molecule in the configuration on the electronic state $|i\rangle$. The ground state vibrational spectrum of molecule $k$ is then given by the time independent Schrödinger equation $h_k^0|\phi_{n_1,n_2}\rangle = \epsilon_{n_1,n_2}|\phi_{n_1,n_2}\rangle$, where $\phi_{n_1,n_2}$ is the 2D vibrational eigenstate with the respective energy $\epsilon_{n_1,n_2}$.

The single molecule cross-sections were computed using the quantum wave packet formalism[17,18], as the half-Fourier transform of the auto-correlation function

$$\sigma_k(\omega, \omega') = \frac{\left|d_{f,c}^k d_{c,0}^k\right|^2}{\pi}$$
$$\times \text{Re} \int_0^\infty dt\, e^{i(\omega-\omega'-\omega_{f0}^k+\epsilon_{0,0}^k+i\Gamma)t}\langle\Psi_k(0)|\Psi_k(t)\rangle, \tag{5}$$

defined by the nuclear wave packets

$$|\Psi_k(0)\rangle = \int_0^\infty dt\, e^{i(\omega-\omega_{i0}^k+\epsilon_{0,0}^k+i\Gamma)t}|\psi_k(t)\rangle,$$
$$|\psi_k(t)\rangle = e^{-ih_c^k t}|\phi_{0,0}^k\rangle, \quad |\Psi_k(t)\rangle = e^{-ih_f^k t}|\Psi_k(0)\rangle,. \tag{6}$$

Here, $d_{i,j}^k$ is the transition dipole moment, at the equilibrium geometry, between the electronic states $i$ and $j$ for the $k$-th molecule. In liquid-phase, the scattered X-ray photons are very likely to be reabsorbed by nearby molecules. We account for this effect by carrying out a self-absorption correction in the same fashion as in previous work[18,28].

For analysis of the RIXS cross-section, we compare to the density of vibrational states

$$\rho_n(\epsilon) = \sum_{k=1}^{64} \sum_{n_1+n_2=n} \Phi\left(\epsilon - \epsilon_{n_1,n_2}^k + \epsilon_{0,0}^k\right), \tag{7}$$

where $\Phi(x) = \exp\left(-x^2/\delta^2\right)/\delta\sqrt{\pi}$, $\delta = 0.01$ eV, $n_1$ and $n_2$ are the vibrational quantum numbers for the stretching modes along the OH bonds in the $k$-th water molecule.

To reconstruct the liquid spectrum $\sigma_{\text{liquid}}(\omega', \omega)$ from the one in gas-phase we convolute the experimental gas-phase spectrum $\sigma_{\text{gas}}(\omega', \omega)$ with the distribution function $\rho(\omega_1' - \omega')$

$$\sigma_{\text{liquid}}(\omega', \omega) \approx \int_{-\infty}^{\infty} \sigma_{\text{gas}}(\omega_1', \omega)\rho(\omega_1' - \omega')d\omega_1' hskip-3', \tag{8}$$

where $\rho(\omega_1' - \omega') = \exp\left(-4\ln 2(\omega_1' - \omega')^2/\gamma^2\right)$. The employed structural inhomogeneous broadening $\gamma(\text{FWHM}) = 0.35$ eV is in reasonable agreement with the 0.45 eV value obtained using molecular dynamics simulations[12].

**Computational details**. For consistency of analysis, the wave packet simulations were performed on the same configuration as our previously classical spectrum simulations[8,9] of liquid water, which was obtained from ab initio MD simulations with periodic boundary conditions in a cubic simulation cell ($a = 12.4170$ Å) in the NVE ensemble in the CPMD programme[42] using the gradient-corrected BLYP functional[43,44] and a 85 Ry kinetic energy cutoff for the plane wave expansion of the Kohn–Sham wave-functions, in combination with a pseudo-potential description. (see refs. [8,9] for further details and results of classical simulations.)

The dissociative $4a_1$ band was modelled with the excited core-hole method (XCH)[35], based on DFT in which the lowest lying core-excited state forms the pre-edge peak in both gas-phase and liquid water. This level of theory has already been shown to accurately reproduce the pre-edge feature of XAS[9,35] of liquid water, which is the main focus of this article. The molecular potentials of and transition dipole moment between the ground state and the lowest O1s core-excited state were computed for each one of the 64 molecules in the sampled configuration. These unrestricted all-electron DFT calculations were performed using the GAPW method in CP2K[37,45,46], employing the BLYP functional[43,44], Ahlrichs-def2-QZVP basis sets[47] and a plane wave cutoff of 300 Ry for the soft part of the density.

**Extraction of confidence interval for OH potentials**. The potential reconstruction procedure was based on a genetic algorithm (GA) and implemented with the help of the deap python library[38]. The individuals were chosen to be a set of parameters, which define the model potential along the OH bond. To obtain the confidence interval for the OH potentials, from the experimental RIXS data, we fitted the parameters ($B, \beta, \alpha, D$) of the modified Morse potential

$$V(R) = V_M(R) + Be^{\beta R}, \quad V_M(R) = D\left(1 - e^{-\alpha R}\right)^2, \quad (9)$$

using the GA[38] to search for potentials that satisfy the constraints defined by Eqs. (1) and (2) and described in the surrounding text.

A random initial population (distribution of potentials) is then generated at the initial step and the evolutionary operations are applied to the population until convergence is reached as it is illustrated in Supplementary Fig. 4. The fitness criterion of our GA was based on the concept of a "vibrational eigenvalue distribution", by which we consider whether a given eigenvalue lies within the width of a given peak $\Delta\varepsilon_m$ in the RIXS spectrum (see Supplementary Fig. 3), we consider the first six peaks ($m = 1, \ldots, 6$) of the experimental RIXS spectrum (which are the most relevant according to our analysis).

**Proof of principle**. To verify the suggested reconstruction technique, it is natural to apply it to the theoretical RIXS spectrum and to compare the obtained potentials to the original ones used to compute RIXS. The results of these test calculations depicted in Supplementary Fig. 5 shows a good agreement between the extracted confidence interval and true distribution of the OH potentials. Clearly, the method will not recover exactly the set of original potentials, but the spread in the distribution is reproduced.

**Local structure classification**. Even though we recognise that hydrogen bonding is not universally defined[21–23,48–52], we employ a definition based on geometrical criteria[24,49]. In our article, we use a geometrical classification of the HB (see Supplementary Fig. 1), which is similar to ref. [24]. For a given molecule, we introduce the oxygen–oxygen radius vector $\mathbf{R}_{OO}$ and the intramolecular OH bond vectors $\mathbf{R}_i$ ($i = 1, 2$) so that the angle $\Theta$ between these vectors is given as $\widehat{\mathbf{R}}_{OO} \cdot \widehat{\mathbf{R}}_i = \cos\Theta$ and the oxygen–oxygen distance is given as $R_{OO} = |\mathbf{R}_{OO}|$. The set of structures, which satisfy the constraint

$$R_{OO} < 3.3 \text{ \AA}, \quad \Theta < 30° \quad (10)$$

only for one OH bond we refer to as D1 (single-donor) structures while the rest of the structures with both OH bonds forming HBs are referred to as D2 (double-donor) structures.

**Choice of transition dipole moment model**. Extraction of local structural information from RIXS measurements of liquids is a widely discussed problem. Within the used MD simulations we obtained a proportion of 20% D1 structure and 80% D2 structures using the geometrical criteria introduced in the previous section. To check whether our combined theoretical/experimental analysis of RIXS can shed light on the relative contribution of these structures, we performed simulations of the RIXS profile for excitation at the pre-edge region, related to the dissociative core-excited state, for the studied configuration using different techniques for computing transition dipole moments. We find that the relative contributions of D1 and D2 structures are very sensitive to the model used to compute the transition dipole moments (see Supplementary Fig. 6).

In an accentuated argument, let us imagine that the transition dipole moments associated with D1 structures are much larger in magnitude than the ones for D2 structures. In such case, we would draw the erroneous conclusion that D1 structures dominate liquid water, when in fact it would only mean that RIXS would not "see" the D2 structures. One should mention that similar questions are raised in XAS studies of liquid water[9,24,25,34]. Supplementary Fig. 6 displays this

problem more accurately. The figure presents the simulation results for three techniques of calculation: One of which exaggerates the transition dipole moment of D1 structures (the half core-hole (HCH) method[37]), and two of which have a more balanced distribution (the full core-hole (FCH)[37] and XCH methods). These three different techniques give the following relative contribution D1/D2 to the total RIXS profile: HCH 86%/14%, FCH 53%/47% and XCH 57%/43%. We see that 86% of D1 contribution in RIXS does not reflect the actual 20% fraction of D1 structures in the configuration obtained from the MD simulations. In spite of this fact, Supplementary Fig. 6 shows that the total RIXS spectra for these three density functional theory (DFT)-based techniques are nearly the same. Thus, we can not draw definite conclusions about the overall local structure in liquid water from RIXS data, but we can clearly conclude that the configurations, which are actually probed at the XAS pre-edge, have a broad distribution of ground potential shapes.

**Role of the single-bond approximation on RIXS**. Now, let us turn our attention to the single-bond approximation usually used in analysis of RIXS of liquid water. In earlier studies[11,16], it is suggested that the vibrational progression seen in RIXS may be understood simply in terms of a Morse potential along a single OH bond, namely the one with a broken HB, which exists only in the D1 subset. As one can see from Supplementary Fig. 1a this approximation $\sigma_{D1}^{sb}$ does not match with the strict theoretical RIXS profile, $\sigma$.

The single-bond approximation, which includes all OH bonds (broken and intact) as well as all structures D1 and D2 gives even worse agreement (Supplementary Fig. 1b). The reason for this discrepancy is that the single bond approximation neglects the fact that the both OH bonds are excited in the scattering process coherently (this is seen in Fig. 3a, c and is accounted for in the strict RIXS profile) resulting in a manifold of mixed excitations.

## Data availability

The computer code and datasets generated and analysed during the current study are available from the corresponding authors on reasonable request.

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

## Acknowledgements

This work was supported by the Swedish Research Council (VR), the Carl Trygger foundation and the Knut and Alice Wallenberg foundation (Grant No. KAW-2013.0020); V.V.C. acknowledges the Conselho Nacional de Desenvolvimento Científico e Tecnológico (CNPq-Brazil); F.G. and V.K. acknowledge the Russian Science Foundation (project 16-12-10109); M.D. and T.S. acknowledge funding from the Swiss National Science Foundation within the D-A-CH programme (SNSF Research Grant 200021L 141325); S.E. and A.F. acknowledge funding from the ERC-ADG-2014—Advanced Investigator Grant no. 669531 EDAX under the Horizon 2020 EU Framework, Programme for Research and Innovation; The work at PSI is supported by the Swiss National Science Foundation through the NCCR MARVEL and the Sinergia project Mott Physics Beyond the Heisenberg Model (MPBH). X.L. acknowledges financial support from the European Community's Seventh Framework Programme (FP7/2007-2013) under Grant Agreement No. 290605 (COFUND: PSI-FELLOW). M.O. and A.F. acknowledge partial funding by the Helmholtz Virtual Institute VI419 "Dynamic Pathways in Multi-dimensional Landscapes"; The calculations were performed on resources provided by the Swedish National Infrastructure for Computing (SNIC).

## Author contributions

The experiment was designed and developed by S.E., A.P., J.N., M.F., M.D., B.K., T.S., X.L., D.M., R.M.J. and A.F. The experiments were performed by J.N., S.E., R.M.J., M.F. and A.P. The data were analysed and interpreted by V.V.C., J.N., A.P., F.G., M.O., V.K., R.C.C. and A.F. The ab initio M.D., potential energy calculations and quantum wave packet simulations were performed by V.V.C., M.O., M.I. and E.E. The manuscript was written by V.V.C., F.G. and M.O. All authors commented on the manuscript. The project was led by M.O., V.V.C. and A.F.

## Additional information

**Competing interests:** The authors declare no competing interests.

