## [Peer Review File · Nature Communications]

Reviewers' comments:

Reviewer #1 (Remarks to the Author):

In the manuscript titled as "Probing hydrogen bond strength in liquid water by resonant inelastic X-ray scattering", the authors performed the x-ray measurement and ab initio molecular dynamics simulation and showed that inelastic resonant x-ray measurement is sensitive to the local structure of water. I found that this is an interesting direction and also a timely topic, but I am not sure whether this technique has more advantages for probing microscopic structure in neat water and in aqueous solution than the other conventional experimental techniques, as the comparison with the different techniques have not been discussed (although the authors mentioned that "RIXS is more sensitive than IR" in the abstract, there is no rich discussion on it.). I suggest the authors to consider the comments given below and enrich the discussion.

About the scholar presentation, some parts of the discussion are not well-explained in the manuscript (see below). Furthermore, I found many careless mistakes. I am quite curious whether all 17 authors read this manuscript carefully and eventually they could not find such mistakes. Apparently, the authors should read the manuscript with greater care. As such, I cannot recommend the editor to accept the manuscript at the present stage, although I think that this manuscript may be potentially published in Nature Communications.

Major comments:

1. I do not understand the claim of shortening of vibrational progression (L53-55): It is not clear why the fluctuation in the OH PEC can shorten the vibrational progression. If my understanding is incorrect, this statement is opposed to the motional narrowing. When the fluctuation is stronger and the time correlation function decays shortly, the power spectrum becomes broader.
2. The connection of point 1 with the statement in L62-63 is not clear.
3. L65-66, I do not understand what the previous empirical analyses are and why this discussion is needed here. If the authors want to explain this, then the authors should explain how the previous empirical analyses are and how this Fig 2c is generated. Otherwise, this whole discussion should be moved to other materials such as Supporting Information.
4. In the end, I could not find any discussion on the comparison of the RIXS data and IR data. The hydrogen bond strength of water has been characterized by IR/Raman spectra, by using the fact that the O-H covalent bond stretch frequency (not only static IR/Raman but also the 2D-IR). It is also necessary to add the quantitative discussion on the comparison of IR/Raman vs. RIXS technique.

Minor comments:

In L38-40, the authors motivate the readers with the current controversy. This is indeed good motivation to introduce a new technique, while citations are not so helpful. First of all, to explain the general topic of a mixture of two structural motives, I guess that a review paper such as Nature Comm. volume 6, 8998 (2015) would be more useful for the readers than a specific one article. Analogously, I think a review paper for explaining the continuum of different H-bonding configuration would be useful. In particular, I was a bit surprised that there is no introduction of vibrational spectroscopy of water; this claim is mostly supported by the IR/Raman/time-resolved IR spectroscopy. I recommend the authors to combine such spectroscopic efforts into this statement (for example, Chem. Rev. 116, 13, 7590 (2016).).

In P 15, at least, the authors should write the condition for ab initio MD simulation. No information can be found for ensemble, no information on cell size, and pressure (if NPT is used). Furthermore, I guess that in the CPMD, the pseudopotential was used, in contrast to the post processing by using CP2K. If so, the authors should mention it.

In L284-285, R and angle are not defined explicitly. Furthermore, the extended Fig 4 "ROO" should be modified.

In L56, Each groups -> Each group

In L63, see insert in Fig. 1(b) -> see inset in Fig. 1(b)

In L74, enhanced transitions dipole moments -> enhanced transition dipole moments

In L78, with their intrinsic spread (?)

In L 116, the very onset (?)

In L 295, remove the unnecessary space

In L 284-285, add period "." after geometrical criteria

in L268, Is ",," appropriate, after "a broken HB"?

Reviewer #2 (Remarks to the Author):

Nature Communications, manuscript NCOMMS-18-33144-T "Probing Hydrogen Bond Strength in Liquid Water by Resonant inelastic X-ray scattering" by V. Vaz de Cruz et al.

The manuscript by V. Vaz de Cruz et al. is devoted to the characterization of gas phase and liquid water by resonant inelastic X-ray scattering (RIXS) and to the interpretation of the experimental data in terms of potential energy curves (PECs), local environment of water molecules, and nuclear dynamics effects. The manuscript combines state-of-the-art experimental data and state-of-the-art theoretical simulations, which both make sense and agree nicely with one another, especially in the part related to the vibrationally resolved RIXS measurements. In my opinion, this part is the major achievement of the manuscript, which makes it suitable for the publication in Nature Communications. At the same time, a certain major revision of the manuscript is necessary as described below.

Specific comments:

- p. 2: there are also "competing conceptions of the local structure of liquid water" in terms of the average number of broken/weak/distorted and intact hydrogen bonds (HBs).

- a recent paper devoted to local structure of liquid water can be considered and cited: Nat. Commun. 4:4150 (2013)

- The vibrationally resolved RIXS spectra for gas phase and liquid water are noticeably different and, consequently, are affected by local structure of HBs. So, these data can be indeed used as "a structural probe in aqueous solution".

- The authors demonstrate that they can separately determine PECs for OH groups involved in weak and strong HBs based on their vibrationally resolved RIXS measurements. Can the authors say something about the average number of broken/weak/distorted and intact HDs in liquid water?

In contrast to the vibrationally resolved RIXS spectra, the results regarding the RIXS channel involving the non-bonding 1b1 lone-pair orbital look less convincing to me. It is already well known (see e.g. ref 8) that core-excitation into the [O1s-1 4a11] state leads to nuclear dynamics on the time scale of the RIXS process, resulting, in particular, to the splitting of the 1b1 peak in the respective XES spectra - a feature of "a purely dynamical origin". Most important, this splitting is observed for both gas phase and liquid water and, therefore, is a purely molecular property and, consequently, not representative of the local structure of HBs. As far as I understand this fact is expressed by the authors as "the

splitting is primary sensitive to the short range part of PEC...". The problem is that these data and the derived conclusions are hardly related to local structure of liquid water but can be interpreted as such. This should be clearly described.

To look at local structure of liquid water, RIXS data corresponding to the higher excitation energies, including the non-resonant case, should be considered. In this case, the splitting of the 1b1 peak is only observed for solid state (ice) and liquid water, in contrast to gas phase water molecules (see e.g. the top experimental spectrum in Figure 4c). Consequently, the splitting of this peak and the branching of individual components in the spectra are characteristic of a certain local structure of HBs and can be used to get specific information about this local structure. In my opinion, this splitting has indeed a purely dynamical origin as has been originally demonstrated in ref 5 (in contrast to the model suggested in ref 4) but it is questionably whether the data taken for the [O1s-1 4a11] state are relevant to support this statement. The respective data are certainly of interest but should be properly discussed to avoid a misinterpretation.

Reviewer 1:

Comment 1: In the manuscript titled as "Probing hydrogen bond strength in liquid water by resonant inelastic X-ray scattering", the authors performed the x-ray measurement and ab initio molecular dynamics simulation and showed that inelastic resonant x-ray measurement is sensitive to the local structure of water. I found that this is an interesting direction and also a timely topic, but I am not sure whether this technique has more advantages for probing microscopic structure in neat water and in aqueous solution than the other conventional experimental techniques, as the comparison with the different techniques have not been discussed (although the authors mentioned that "RIXS is more sensitive than IR" in the abstract, there is no rich discussion on it.). I suggest the authors to consider the comments given below and enrich the discussion.

We agree that our comparison with alternative experimental techniques is rather schematic. Hence, we extended the comparison to (and related discussion of) IR spectroscopy throughout the article. The fact that quasi-elastic RIXS and IR absorption and IR Raman spectroscopies probe the vibrations in ground state makes these techniques complementary to each other. The strong distinction between the RIXS and IR techniques is the number of vibrational states "seen" in the spectra. To make this comparison more clear we added a new figure (Fig. 2) which clearly shows that quasi-elastic RIXS of liquid water probes many more vibrational O-H stretching resonances than accessible in the IR spectrum. More precisely, the IR experimental data display a strong $\nu = 1$ OH stretch band and more than two orders of magnitude weaker dipole forbidden $\nu = 2$ and $\nu = 3$ resonances (see Ref.[30]). The long vibrational progression in RIXS allows us to probe more directly the long-range portion of the OH potential, which is strongly influenced by the hydrogen bond (HB). In contrast, the modern very accurate IR spectroscopy is focused mainly on the deep and detailed analysis of the shape of the dominant 0-1 OH stretching peak in IR spectra which allows to gain important information, for example about local structure and vibrational energy redistribution. Notice, that the sensitivity of IR spectroscopy to a large extent stems from the strong dependence of the IR intensity of the $\nu = 1$ OH stretch on the hydrogen bond environment. This gives complementary sensitivity to hydrogen bonding, which not simply a measure of the broad distribution of vibrational oscillators. These points being crucial for understanding the advantages of different techniques in studies of liquid are enhanced in our modified text. A presentation of IR spectroscopy of liquid water and aqueous solutions with review references is now given in the introduction and the relation to RIXS is discussed across the article, including also time-resolved pump-probe IR and 2DIR.

Comment 2: About the scholar presentation, some parts of the discussion are not well-explained in the manuscript (see below). Furthermore, I found many careless mistakes. I am quite curious whether all 17 authors read this manuscript carefully and eventually they could not find such mistakes. Apparently, the authors should read the manuscript with greater care. As such, I cannot recommend the editor to accept the manuscript at the present stage, although I think that this manuscript may be potentially published in Nature Communications.

We appreciate the feedback from the reviewer and we have checked carefully the text of the article and tried to fix all misprints. Following to referee's suggestion, we improved the presentation of the article to make it clearer and to

address the raised concerns. We hope that the editor and the referees will find the new version suitable for publication on Nature Communications.

Comment 3: I do not understand the claim of shortening of vibrational progression (L53-55): It is not clear why the fluctuation in the OH PEC can shorten the vibrational progression. If my understanding is incorrect, this statement is opposed to the motional narrowing. When the fluctuation is stronger and the time correlation function decays shortly, the power spectrum becomes broader.

This is a valid and insightful comment. The motional narrowing caused by the fluctuations in the environment can affect the spectral shape of individual vibrational peaks of quasi-elastic RIXS similarly to IR absorption. The reason for this is that both RIXS and IR end up in the same final state. This dynamical effect is neglected here because we use a static environment (see Methods). To justify the static approximation used here, it is worth noting that the broadening of a vibrational resonance has two representative limits defined by the dimensionless parameter $\Delta\omega\tau_c$, where $\Delta\omega$ is the variance of frequency fluctuation in the liquid, while τ_c is the decay time of the frequency fluctuation correlation function. In the regime of slow modulation (static regime) $\Delta\omega\tau_c \gg 1$ the line width is large and is given by inhomogeneous broadening. The regime of motional narrowing occurs in the opposite case of fast modulation $\Delta\omega\tau_c \ll 1$ in which the line width is defined by the homogeneous broadening. According to 2D IR spectroscopy $\tau_c \approx 176$ fs for water (see Ref.[32]). MD simulations (see Ref.[31]) have shown that bath fluctuations reduce the spectral width of the main OH IR peak (n=1) around 30% for a value of $\Delta\omega\tau_c \sim 1$. In the case of RIXS both experiment and simulations (Fig. 1b) show that $\Delta\omega$ grows rapidly on the way to higher vibrational resonances (main focus of our study) where $\Delta\omega\tau_c > 1$ and the regime of motional narrowing is switched to the static regime.

Thus strong fluctuations of long range of the OH potential from structure to structure, and hence of OH frequency, results in the overlap of the peak positions of high vibrational levels, which smears the vibrational structure in quasi-elastic RIXS as one can see both from theory and experiment (Fig.1). The related discussion was incorporated into the main text to clarify these points.

Comment 4: The connection of point 1 with the statement in L62-63 is not clear.

We have attempted to elaborate the presentation to clarify the relation between vibrational density of states and features of the RIXS spectrum; Namely, the shortening of the progression and the increased peak width. In particular we replaced the discussed sentence by a new one: "Thus, the overlap of the partial density of states ρ_n , and the related increase in peak width (see inset in Fig. 1b), qualitatively explains the shortening of the spectrum".

Comment 5: L65-66, I do not understand what the previous empirical analyses are and why this discussion is needed here. If the authors want to explain this, then the authors should explain how the previous empirical analyses are and how this Fig 2c is generated. Otherwise, this whole discussion should be moved to other materials such as Supporting Information.

To answer this comment we wrote in the Introduction: "Earlier, the long vibrational progression in quasi-elastic RIXS has been empirically analysed; Either attributed to highly weakened/broken donating HBs selected by the pre-edge core-excitation in the framework of a single bond approximation [16]. Or assigned to symmetric and antisymmetric normal modes and to OH vibrations in a broken-bond molecule [11]. However, a water molecule has in general two non-equivalent OH bonds, due to the asymmetric surroundings. This necessitates a strict coherent treatment of both oscillators as carried out in our simulations presented below." We clarified this point in detail, including an explanation of the discussed Fig. 3c (former Fig. 2c) in Sec: "Role of asymmetric bonds".

Comment 6: In the end, I could not find any discussion on the comparison of the RIXS data and IR data. The hydrogen bond strength of water has been characterised by IR/Raman spectra, by using the fact that the O-H covalent bond stretch frequency (not only static IR/Raman but also the 2D-IR). It is also necessary to add the quantitative discussion on the comparison of IR/Raman vs. RIXS technique.

As mentioned above, we have added key references and a discussion about IR of water, 2DIR and pump-probe IR also in the introduction. Moreover, we paid special attention (with a related reference) to the crucial role of the 2D-IR time-resolved spectroscopy to extract such an important parameter as the correlation time which defines the border between motional narrowing and static regimes (see also our Reply to Comments 1 and 3 of referee 1). We added also a comparison between IR and RIXS data (see new Fig.2 and related discussion).

Comment 7: Minor comments: In L38-40, the authors motivate the readers with the current controversy. This is indeed good motivation to introduce a new technique, while citations are not so helpful. First of all, to explain the general topic of a mixture of two structural motives, I guess that a review paper such as Nature Comm. volume 6, 8998 (2015) would be more useful for the readers than a specific one article. Analogously, I think a review paper for explaining the

continuum of different H-bonding configuration would be useful. In particular, I was a bit surprised that there is no introduction of vibrational spectroscopy of water; this claim is mostly supported by the IR/Raman/time-resolved IR spectroscopy. I recommend the authors to combine such spectroscopic efforts into this statement (for example, Chem. Rev. 116, 13, 7590 (2016)).

We are grateful to the reviewer for this insightful comment and several suggestions. We have answered partly this comment in our Replies on Comments 1 and 6 together with corresponding changes in the text. Now both articles Nature Comm. 6, 8998 (2015) and Chem. Rev. 116, 13, 7590 (2016) are cited in added discussion of extraction of structural information from IR and X-ray experiments (see also our reply to Comment 1 of the referee 2).

Comment 8: In P 15, at least, the authors should write the condition for ab initio MD simulation. No information can be found for ensemble, no information on cell size, and pressure (if NPT is used). Furthermore, I guess that in the CPMD, the pseudopotential was used, in contrast to the post processing by using CP2K. If so, the authors should mention it.

Together with reference to previous work, the essential details of the *ab initio* MD simulation are stated in Sec. "Computational details".

Comment 9: In L284-285, R and angle are not defined explicitly. Furthermore, the extended Fig 4 ?ROO? should be modified.

We have made the geometrical hydrogen bond classification used clear by explicitly defining the angle Θ and the oxygen-oxygen distance R_{OO} in the text (Sec. "Local structure classification"). We have also modified Fig.4 of SI according to the suggestion of the referee.

Comment 10: In L56, Each groups -> Each group In L63, see insert in Fig. 1(b) -> see inset in Fig. 1(b) In L74, enhanced transitions dipole moments -> enhanced transition dipole moments In L78, with their intrinsic spread (?) In L 116, the very onset (?) In L 295, remove the unnecessary space In L 284-285, add period ?.? after ge-

ometrical criteria in L268, Is ?,? appropriate, after ?a broken HB??

We have fixed the typos pointed out by the referee. For clarification “the very onset of the splitting” was rephrased to “the emergence of the splitting at the pre-edge resonance...”.

Reviewer 2:

The manuscript by V. Vaz da Cruz et al. is devoted to the characterisation of gas phase and liquid water by resonant inelastic X-ray scattering (RIXS) and to the interpretation of the experimental data in terms of potential energy curves (PECs), local environment of water molecules, and nuclear dynamics effects. The manuscript combines state-of-the-art experimental data and state-of-the-art theoretical simulations, which both make sense and agree nicely with one another, especially in the part related to the vibrationally resolved RIXS measurements. In my opinion, this part is the major achievement of the manuscript, which makes it suitable for the publication in Nature Communications. At the same time, a certain major revision of the manuscript is necessary as described below

We are deeply grateful to the reviewer for considering our work, for recognising the achievements in the study and for giving valuable feedback on how to improve our manuscript. We describe below how we attended to the comments.

Comment 1: - p. 2: there are also "competing conceptions of the local structure of liquid water" in terms of the average number of broken/weak/distorted and intact hydrogen bonds (HBs).

We fully agree that averaged number of HB per molecule in liquid water is one of the major parameters which scientific community tries to extract from the measurements. Motivated by this insightful comment a discussion was added to the Introduction about the complexity of extracting HB statistics of liquids from core-level spectroscopy due to the intrinsic uncertainty in the calculation of transition dipole moments of highly excited states of large systems. We have also motivated our new approach of investigating the local HB environment by analysing RIXS of liquid water to derive local variations in the ground state potential, which is a central result in our study. Thereby, we could characterise the variations in the local OH PEC for strongly and weakly hydrogen bonded situations.

Comment 2: - a recent paper devoted to local structure of liquid water can be considered and cited: Nat. Commun. 4:1450 (2013)

The reviewer refers to an interesting article Nat. Commun. 4:1450 (2013), which reconciles the articles by Wernet *et al* [25] and Chen *et al* [26]. The results of this article is that small asymmetric (D1) fraction is responsible for the formation of the pre-edge peak. Here we face the question of how accurate the calculation of the transition dipoles moments of core excitation d_{c0} is. The authors of Nat. Commun. 4:1450 (2013) article use the half-core-hole (HCH) methods to compute the intensity of XAS (cross-section $\propto d_{c0}^2$). In our article we use three different methods (XCH, FCH and HCH) to compute the intensity of quasi-elastic RIXS (cross-section $\propto d_{c0}^4$). The MD simulation of the cited work gives approximately a 20% fraction of single donor structures (D1). However, the simulations of d_{c0} based on the HCH method yield significantly larger transition dipole moments for D1 structures than for D2 structures in comparison with FCH and XCH techniques. This explains why, contrary to the FCH and XCH methods, the HCH technique predicts a considerably larger signal from D1 structures in the quasi-elastic RIXS (Fig. 5 from SI) and in pre-edge peak of XAS in spite of the fact that the fraction of D2 structures ($\approx 80\%$) is dominant in MD calculations. Our results based on the HCH are in agreement with the article Nat. Commun. 4:1450 (2013). We now cite their results in the text (shown in our Reply on Comment 1 of the referee 2).

Comment 3: The vibrationally resolved RIXS spectra for gas phase and liquid water are noticeably different and, consequently, are affected by local structure of HBs. So, these data can be indeed used as "a structural probe in aqueous solution".

We are grateful that the reviewer recognises our conclusion of the information content and the value of analysing the RIXS spectra to extract information about the hydrogen bonding environment.

Comment 4: The authors demonstrate that they can separately determine PECs for OH groups involved in weak and strong HBs based on their vibrationally resolved RIXS measurements. Can the authors say something about the average number of broken/weak/distorted and intact HDs in liquid water?

As seen in Fig. 3 (b and c), we notice clear trends distinguishing different HB environment; Namely, D1 and D2 configurations. Based on these trends we constructed confidence intervals consistent with the experimental spectrum. However, the extraction of meaningful quantitative ratios of the existing local environments from the RIXS data alone is in principle an unattainable task,

since it requires knowledge of intensities associated with each transition in each local environment. Instead, we assume the RIXS peaks define the intervals wherein the vibrational eigenvalues are allowed to exist and then explore the parameter space of a model potential using genetic algorithm searches in order to estimate the distribution of possible OH stretching potential energy curves. Hence the strongest conclusion which can be drawn from this analysis is that a broad distribution of different configurations are probed at the O1s X-ray absorption pre-edge instead of a very narrow one as previously suggested. This conclusion was explicitly stated in the main text.

Comment 5: In contrast to the vibrationally resolved RIXS spectra, the results regarding the RIXS channel involving the non-bonding 1b1 lone-pair orbital look less convincing to me. It is already well known (see e.g. ref 8) that core-excitation into the [O1s-1 4a11] state leads to nuclear dynamics on the time scale of the RIXS process, resulting, in particular, to the splitting of the 1b1 peak in the respective XES spectra - a feature of "a purely dynamical origin". Most important, this splitting is observed for both gas phase and liquid water and, therefore, is a purely molecular property and, consequently, not representative of the local structure of HBs. As far as I understand this fact is expressed by the authors as "the splitting is primary sensitive to the short range part of PEC...". The problem is that these data and the derived conclusions are hardly related to local structure of liquid water but can be interpreted as such. This should be clearly described.

The discussion of the splitting in the electronically inelastic RIXS (1b1 splitting) is included because two qualitatively different explanations of this splitting exist. One is based on the dynamical origin of the splitting while the second attributes this splitting to two structure motives.

Although we agree that the role of nuclear dynamics in the 1b1 splitting had been established in RIXS experiment by the nicely pronounced isotope effect, a clear microscopical explanation was absent. Also, simultaneously with the dynamical origin of the of the 1b1 splitting this splitting has been attributed to the two different structure motives where the nuclear dynamics was invoked to explain the isotope effect under post-edge excitation. A natural question arises: "Can the discussed splitting deliver information about local structure?" This is the main motivation of our quantum investigation of this problem presented in Sec. "Dynamical origin of the 1b1 splitting". In this section we show that the formation of this splitting happens in short-range part of the potential which is very weakly affected by the HB environment and, hence, supports the dynamical interpretation. This explains the very close spectral shapes of the 1b1 doublet in gas and liquid. This comment of the referee is accounted in modified Introduction and Sec."Dynamical origin of the 1b1 splitting".

Comment 6: To look at local structure of liquid water, RIXS data corresponding to the higher excitation energies, including the non-resonant case, should be considered. In this case, the splitting of the $1b_1$ peak is only observed for solid state (ice) and liquid water, in contrast to gas phase water molecules (see e.g. the top experimental spectrum in Figure 4c). Consequently, the splitting of this peak and the branching of individual components in the spectra are characteristic of a certain local structure of HBs and can be used to get specific information about this local structure. In my opinion, this splitting has indeed a purely dynamical origin as has been originally demonstrated in ref 5 (in contrast to the model suggested in ref 4) but it is questionable whether the data taken for the [O1s-1 4a11] state are relevant to support this statement. The respective data are certainly of interest but should be properly discussed to avoid a misinterpretation.

We agree that our discussion of the underlying physics in the excitation energy dependence of the $1b_1$ splitting is not complete.

Despite that the detailed mechanism of the discussed splitting in the main- and post-edge regions (and for non-resonant core-excitations) is beyond the scope of our article, it deserves a special comment. According to DFT-based MD simulations, the doublet in the region of the $1b_1$ peak for non-resonant excitations is related to the $3a_1$ and $1b_1$ levels, which approach each other in the course of the OH bond elongation in the core-ionised state, which as confirmed by the simulations is dissociative in the local hydrogen bond environment in liquid water [8,9]. It is important to notice that this splitting for higher excitation energies is absent in the gas phase [10] where (contrary to liquid phase) the OH potential is bound [9]. Following the referee's suggestion we added this discussion in Sec."Dynamical origin of the $1b_1$ splitting".

On behalf of the author group,

Michael Odelius

REVIEWERS' COMMENTS:

Reviewer #1 (Remarks to the Author):

The manuscript was revised and I think that the current manuscript has much better scholar presentation. I am thus happy to suggest the editor to accept the manuscript.

Reviewer #2 (Remarks to the Author):

I have already commented on the meaning and the importance of the results presented in the paper in my Report to the original version. The paper has been improved significantly upon the Revision and I am mostly satisfied with the changes.

I think that the paper can be accepted as is.

I have just one specific comment, which the author can consider or deal with if they want. I found it interesting that the theoretical curves for both liquid and gas do not reproduce properly the experimental ones for the highest excitation energies in Figures 5b and 5c. The amplitude of the "mol." peak is underestimated for liquid and overestimated for gas.

We thank the editor and the reviewers for their efforts and highly appreciate their constructive comments on our manuscript. As suggested by the editor, we have addressed the final comments by the second reviewer and below we present our response and action to the comment.

Reviewer 2:

Comment: I have just one specific comment, which the author can consider or deal with if they want. I found it interesting that the theoretical curves for both liquid and gas do not reproduce properly the experimental ones for the highest excitation energies in Figures 5b and 5c. The amplitude of the "mol." peak is underestimated for liquid and overestimated for gas.

Our revision in red in the manuscript is:

- Fig. 5c shows some disagreement between theory and experiment for the gas phase RIXS spectra. This is due to the limited accuracy of the calculated OH potentials for the core-excited state and the final states. Environment-dependent fluctuations in emission energy near equilibrium can be characterised by a distribution function $\rho(\omega'_1 - \omega')$. Thus, we can **approximately** reconstruct the liquid spectrum $\sigma_{\text{liquid}}(\omega', \omega)$ by convolution of the experimental gas-phase spectrum $\sigma_{\text{gas}}(\omega', \omega)$ with $\rho(\omega'_1 - \omega')$ (see Methods). In Fig. 5, the reconstructed spectrum $\sigma_{\text{liquid}}(\omega', \omega)$ is shown to be in good agreement with the experimental spectrum of liquid water. **The employed convolution ignores slight variation of the short range part of OH potentials for different structures. This is the main reason for the remaining disagreement between theory and experiment seen in Fig. 5b.**

On behalf of the author group,

Michael Odelius